



# Upgrade of LSA-SAF Meteosat Second Generation daily surface albedo (MDAL) retrieval algorithm incorporating aerosol correction and other improvements

Daniel Juncu[1], Xavier Ceamanos[1], Isabel F. Trigo[2], Sandra Gomes[2], and Sandra C. Freitas[2,3]

[1]CNRM, Météo-France, CNRS, Université de Toulouse, Toulouse, France

[2]Instituto Português do Mar e da Atmosfera (IPMA), Lisboa, Portugal

[3]Deimos Engenharia, Lisboa, Portugal

**Correspondence:** Xavier Ceamanos (xavier.ceamanos@meteo.fr), Daniel Juncu (daniel.juncu@meteo.fr)

**Abstract.** MDAL is the operational Meteosat Second Generation (MSG) derived daily surface albedo product that is generated and disseminated in near real time by the EUMETSAT Satellite Application Facility for Land Surface Analysis (LSA-SAF) since 2005. We propose and evaluate an update to the MDAL retrieval algorithm which introduces the accounting for aerosol effects, as well as other scientific developments: pre-processing recalibration of radiances acquired by the SEVIRI instrument

aboard MSG and improved coefficients for atmospheric correction as well as for albedo conversion from narrow- to broad-band. We compare the performance of MDAL broad-band albedos pre- and post upgrade with respect to two types of reference data: EPS-Metop based 10-day albedo product ETAL (which was found to be comparable to MODIS-based albedo in terms of accuracy) and albedo derived from in-situ flux measurements acquired by ground stations. For the comparison to ETAL — based on differences over the whole coverage area of SEVIRI — we see a reduction of average white-sky albedo mean

bias error (MBE) from $-0.02$ to negligible levels ($< 0.001$), and a reduction of average mean absolute error (MAE) from 0.034 to 0.026 (–24 %). Improvements can be seen for black-sky albedo as well, albeit less pronounced (14 % reduction in MAE). Further analysis distinguishing individual seasons, regions, and land covers show that performance changes have spatial and temporal dependence: for white-sky albedo we see improvements over almost all regions and seasons relative to ETAL, except for Eurasia in winter; resolved by land cover we see a similar effect with improvements for all types for all

seasons except winter, where some types exhibit slightly worse results (crop-, grass- and shrublands). For black-sky albedo we similarly see improvements for all seasons when averaged over the full dataset, although sub-regions exhibit clear seasonal dependence: performance of the upgraded MDAL version is generally diminished in local winter but better in local summer. The comparison against in-situ observations is less conclusive due to the well known problem of spatial representativeness of near-ground observations with respect to satellite pixel footprint sizes. Considering all evidence presented in this study, the

updated algorithm version is considered to be able to deliver a valuable improvement of the operational MDAL product.





# 1 Introduction

Land surface albedo, the ratio of upward- to downward solar radiation, is a key component of the Earth's surface radiation budget relevant in various research and operational fields and has been declared an essential climate variable by GCOS (the Global Climate Observing System). It is needed, for example, in physical models of the atmosphere that incorporate energy balance (e.g. weather prediction or climate models), or can be used for land cover monitoring, to track processes such as deforestation and desertification (e.g. Dirmeyer and Shukla, 1994; Becerril-Piña et al., 2016; Wu et al., 2019). Satellite based land surface albedo retrievals, in particular, play a crucial role in this context as they are available with high spatial coverage at continental or global scale.

The European Organisation for the Exploitation of Meteorological Satellites, EUMETSAT, operates two satellite missions that are used as a basis for land surface albedo products, the geostationary *Meteosat* mission (consisting of a series of satellites with the same name) and the polar orbiting *EUMETSAT Polar System* (EPS; comprising of the Metop satellite series). For exploiting its missions' observations, EUMETSAT oversees a network of *Satellite Application Facilities* (SAF's) of which the Satellite Application Facility for Land Surface Analysis (LSA-SAF, Trigo et al., 2011) is responsible for the generation of surface albedo as part of a wide range of land surface variables.

In this capacity, the LSA-SAF generates a Meteosat albedo product, MDAL, which is a near real-time product disseminated daily with a timeliness of three hours. The product is based on multi-spectral radiance measurements made by the *Spinning Enhanced Visible and InfraRed Imager*, SEVIRI, mounted on the Meteosat Second Generation (MSG) satellite series (Meteosat satellites 8, 9, 10 and 11). As MSG is a satellite system in geostationary orbit, the satellites' location relative to Earth (and the field of view of the Earth's surface) do not change over time, yielding two static areas of coverage: the primary 0° *Service* (currently served by Meteosat-11, see Figure 1) covers an area centered at 0° longitude and 0° latitude (which includes all of Europe and Africa), and *the Indian Ocean Data Coverage Service* (IODC), which is served by Meteosat-8 located over the equator at 41° East (an IODC albedo product is currently generated and made available on a best-effort basis). Both of these fields of view cover a "disk"-like area extending about 80° in either direction from the sub-satellite point. MDAL is used for various research purposes (e.g., Ghilain et al., 2011; Cedilnik et al., 2012; García et al., 2013), as well as as the base for several derived operational LSA-SAF products, such as evapotranspiration and vegetation parameters (Arboleda et al., 2017; García-Haro et al., 2005). The original MDAL product (MDAL v1 hereafter) has been available since 2005, with minor changes to the retrieval algorithm.

In this paper, we propose an update of the MDAL retrieval process (MDAL v2 hereafter), which introduces accounting for aerosol effects as well as several further changes aimed at improving albedo estimates. For validation, we compare the performance of MDAL v2 to that of v1 using two different reference data sets: Satellite derived EPS ten-day albedo (ETAL), and ground station measurements from different locations in Africa and Europe, in both cases using data spanning one full year to account for seasonal variations. The objective of this study is to use this validation to provide justification and scientific



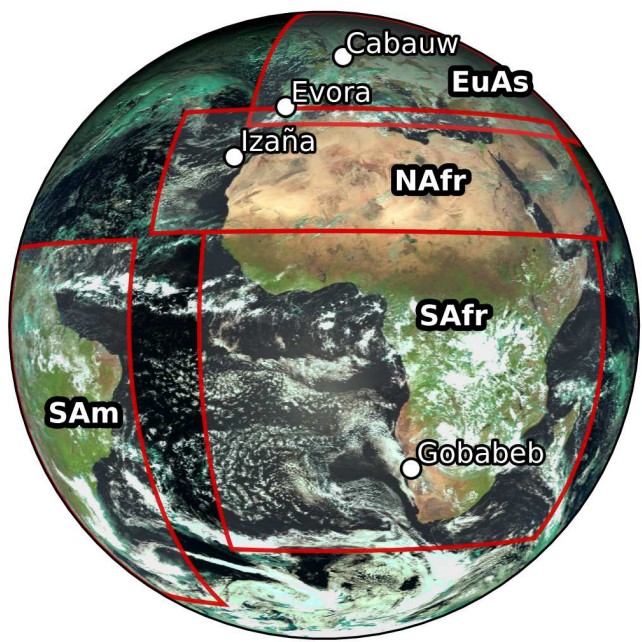

**Figure 1.** False color MSG SEVIRI image showing the region of primary coverage ("full disk"). Red boxes outline the sub-regions used for analysis (EuAs: Eurasia; NAfr: Northern Africa; SAfr: Sub-Saharan Africa; SAm: South America); white circles indicate locations of in-situ stations.

background for the update of the retrieval algorithm in the LSA-SAF operational processing chain, as well as to inform users about the extent of which it would cause changes to MDAL results.

## 2 Albedo retrieval algorithm

### 2.1 MDAL v1: current retrieval algorithm

MDAL is an operational daily surface albedo product based on SEVIRI images of the section of Earth's surface seen from the MSG orbital position in 15 minutes intervals, with a ground resolution of 3 km x 3 km at nadir, with pixel sizes increasing from the centre. MDAL is based on a mature retrieval algorithm described in Geiger et al. (2008), which can be summarized as follows:

1. Ingest SEVIRI radiances for visible and near-infrared channels (central wavelength: VIS06: $0.6\,\mu$m; VIS08: $0.8\,\mu$m; NIR16: $1.6\,\mu$m);

2. mask cloudy pixels (based on cloud mask provided by the NWC SAF);





3. correct radiances for atmospheric effects using SMAC (Simplified Method for Atmospheric Correction, Rahman and Dedieu, 1994) to obtain top-of-canopy (TOC) reflectances. SMAC performs an efficient correction for the different interactions between solar radiation and atmospheric constituents such as gases and, if included, aerosols;

4. accumulate TOC reflectances over one day, then calculate parameters of the bidirectional reflectance distribution function (BRDF) for that day;

5. perform angular integration of BRDF to obtain spectral albedos, both white-sky albedo (also bi-hemispherical, BH, reflectance) and black-sky albedo (or directional-hemispherical, DH, reflectance);

6. perform spectral integration to obtain broadband albedo (full solar spectrum, visible- and near-infrared spectrum; see Appendix A3).

Additionally, the algorithm employs a Kalman filter to propagate the BRDF parameters in time to be used as a priori information in future retrievals. This step helps to avoid data gaps and to stabilize the results. MDAL has previously been validated against the MODIS based surface albedo product (MODIS is the imaging spectroradiometer onboard NASA's *Terra* and *Aqua* satellites; the correspondant albedo retrieval algorithm is developed by the MODIS Land Science Team), with satisfactory results (Carrer et al., 2010). The MDAL product suite contains several albedo variables, consisting of broadband albedo (obtained in step 5 above) as well as single-channel albedos (obtained in step 4), and variances for each dataset. Our analysis is focused on the two full solar spectrum albedo products, white-sky albedo (product variable AL-BB-BH) and black-sky albedo (product variable AL-BB-DH), which are the MDAL variables that are required by most land surface- and weather forecast models.

## 2.2 Proposed upgrade to MDAL v2

One shortcoming in MDAL v1 is the lack of correction for aerosol effects, i.e. their contribution to attenuation (scattering and absorption) of incoming solar radiation in the atmosphere, thus ignoring a major physical effect. Simply speaking, their omission results in attenuation effects that take place in the atmosphere being misattributed to the surface, leading to a distortion in estimated surface reflectance. This distortion can take the form of underestimation over bright surfaces or overestimation over dark surfaces; essentially depending on whether surface reflectivity is higher or lower than aerosol reflectivity. The reason for the previous exclusion of aerosol correction was the difficulty in obtaining acceptable processing results when they were integrated, mainly due to the lack of reliable input data on aerosol optical depth (AOD, the quantity used to measure atmospheric aerosol content). Indeed, some tests using climatological (i.e., long term averaged) AOD values showed a degraded quality of the estimated albedo compared to no aerosol compensation. Because the situation regarding available aerosol data has significantly improved in recent years, however, this shortcoming can now be addressed.

It is notable that the previous omission of aerosol effects in the processing did not result in recognizable flaws in albedo estimates. We believe that this may be due to an interplay of the aerosol omission and other sources of error (possibly attenuating each other's effects), which is why we decided on a comprehensive update, adding the effect of aerosols, but further addressing





three other issues with potentially significant impact on results by adding SEVIRI radiance bias correction and updating SMAC
as well as narrow- to broadband conversion coefficients. In detail:

1. **Incorporation of aerosol effects.** Thanks to work done within the framework of the Copernicus Atmosphere Monitoring
   Service (CAMS), reliable climatological AOD estimates have been made available, since 2017, as part of the CAMS
   reanalysis suite (CAMSRA, Inness et al., 2019). In MDAL v2 we incorporate monthly AOD based on averages of
CAMSRA AOD from the time period of 2003 to 2012, linearly interpolated for each day (see Figure 2 as an example).
   We chose reanalysis AOD over AOD forecasts based on the findings of Ceamanos et al. (2014): Their experiments with
   respect to retrieval of LSA-SAF downwelling surface shortwave flux (DSSF) suggest the reanalysis AOD to be preferable
   due to biases introduced into DSSF estimates when using CAMS AOD forecasts. The capability for incorporating aerosol
   effects in the MDAL processing chain already existed as part of SMAC, only previously AOD had been set to be zero
for all pixels.

2. **SEVIRI bias correction.** Meirink et al. (2013) investigated the bias of SEVIRI shortwave channels with respect to com-
   parable bands of MODIS based on regression analysis of collocated near-nadir reflectance measurements. They found
   negative biases for the two visible channels, and a positive bias for the near infrared channel. Based on their findings the
   authors publish calibration slopes (https://msgcpp.knmi.nl/solar-channel-calibration.html, accessed 17/02/2022) which
can be used to calculate recalibration coefficients for a given point in time. We extract the coefficients for the time in-
   terval used for validation (Nov 2020 to Oct 2021) and apply them in MDAL v2 to adjust the input radiance values.
   The resulting adjustments are, for each channel: VIS06: $+10\,\%$; VIS08: $+6\,\%$; NIR16: $-4\,\%$. We expect this change to
   generally lead to increased albedo estimates. According to Ceamanos et al. (2021), including bias correction provided
   improved AOD retrievals from SEVIRI.

3. **Updated SMAC coefficients.** SMAC is a simplified method to compensate satellite data for atmospheric effects based
   on the more complex, but computationally prohibitive 6S radiative transfer algorithm (Vermote et al., 1997). For SMAC
   to work, 6S results have to be calculated only once (for a given sensor) and the results are used to fit the equations
   used for parametrization in SMAC. The coefficients obtained as a result of this fitting can then be used for the SMAC
   correction of individual images, using the same equations. A vector version of 6S, 6SV1, was developed by Kotchenova
et al. (2006), enabling accounting for the effect of radiation polarization, among other improvements. MDAL v2 uses
   the re-calculated SMAC coefficients based on 6SV1, while MDAL v1 uses 6S.

4. **Updated narrow- to broadband conversion coefficients.** The narrow- to broadband conversion in the MDAL retrieval
   algorithm involves a set of conversion coefficients which are calculated based on regression analysis using synthetic
   spectral reflectances of different surface types (generated from Advanced Spaceborne Thermal Emission Reflection
Radiometer, ASTER, data), fitted by the SAIL (Scattering by Arbitrary Inclined Leaves) radiative transfer model (for
   more details see section 2.5.2 in Carrer et al., 2021). The database of synthetic spectral reflectances has been updated to

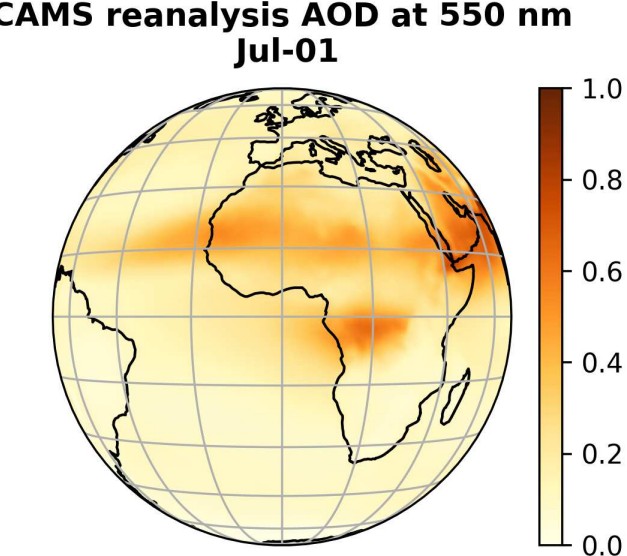

**Figure 2.** Aerosol optical depth (AOD) used in MDAL v2 processing. Values for July 1st based on CAMS reanalysis (monthly values, linearly interpolated to obtain daily values).

include a more exhaustive set of vegetation and bare soil surfaces and has been used to obtain conversion coefficients for MDAL v2 (values of coefficients are given in Appendix, Table A2).

Furthermore, we found that the incorporation of aerosol correction led to additional problems stemming from the limitations

of SMAC (limited reliability at high solar- and view zenith angles; see Rahman and Dedieu, 1994; Proud et al., 2010). Those limitations are amplified, in particular in the outer region of the SEVIRI disk, leading to unrealistic values with errors increasing as a function of view zenith angle, solar zenith angle and aerosol optical depth. We were able to mitigate those effects by limiting the solar zenith angles that are taken into account for each retrieval, and by discarding obviously unphysical values after the atmospheric correction step. More precisely, we now:

– exclude observations with solar zenith angles above $80°$ (instead of $85°$ as used before),

    – discard negative TOC reflectances after atmospheric correction, and

    – discard TOC reflectances above 1.5 (previous upper limit was 3.2768, for numerical reasons).

It should be noted that the reduction of solar zenith angles from $85°$ to $80°$ leads to a (minor) loss of coverage at high latitudes in winter.





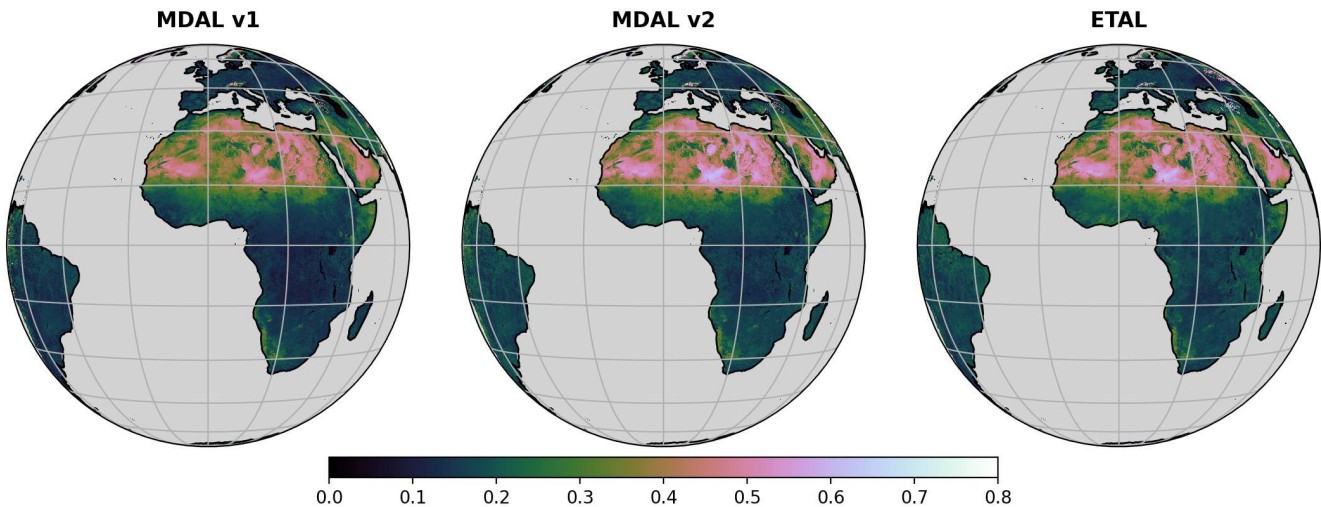

**Figure 3.** Broadband bi-directional hemispherical (BB-BH) Albedo of: MDAL v1, MDAL v2 and resampled ETAL. Date: 15/04/2021.

# 3  Data

## 3.1  Data to evaluate: MDAL

For the analysis we use a full year of MDAL data, both for v1 and v2, spanning from 1-Nov-2020 to 31-October-2021. The data covers the full disk observed by MSG primary coverage (Meteosat-11 centered at $0°/0°$, coverage to approximately $80°$ N, S, E, W). Both experiments are based on the retrieval algorithm outlined in Section 2.1, with MDAL v2 incorporating the changes described in Section 2.2. Both MDAL v1 and v2 are initialized with the same starting conditions, i.e. the same a priori BRDF parameters ingested by the Kalman filter. This implies that during the first weeks, the MDAL v2 results will have a bias towards MDAL v1 due to the memory of the Kalman filter. This effect disappears with time.

The data has a temporal resolution of one day, the grid spacing is 3 km at Nadir and increases towards the disk's edge. Albedo maps of both MDAL versions are shown in Figure 3 (left and center).

## 3.2  Reference data 1: ETAL

ETAL, short for EPS ten-day surface albedo, is a global near real-time operational albedo product generated by the LSA-SAF since 2015. ETAL is based on observations made by the AVHRR instrument aboard the primary Metop satellite (currently Metop-B) which is part of the EUMETSAT Polar System (EPS). The Metop satellites are polar orbiting at an altitude of around 830 km and its AVHRR sensor provides images with a ground resolution of approximately 1 km x 1 km. The data are collected into daily global images, which in turn are accumulated over a 20-day composite window (the longer time window when compared to MDAL is necessary to compensate the lower temporal resolution — one per day — of polar orbiting satellites


|  | SEVIRI | | AVHRR | |
| --- | --- | --- | --- | --- |
| Channel name (SEVIRI) | Center [nm] | Width [nm] | Center [nm] | Width [nm] |
| VIS06 | 635 | 150 | 630 | 100 |
| VIS08 | 810 | 140 | 865 | 275 |
| NIR16 | 1600 | 280 | 1610 | 60 |

**Table 1.** Spectral characteristics of SEVIRI and AVHRR channels.

compared to geostationary satellites). A global albedo product is generated approximately every 10 days (the 5th, 15th and 25th of each month), based on the preceding 20-day window. Except for the differences in temporal composition, the ETAL retrieval algorithm is generally the same as for MDAL. Three of the four proposed changes to the MDAL retrieval algorithm

(see Section 2.2; all except SEVIRI bias correction as it does not apply to AVHRR) are already implemented for ETAL, taken into account differences in sensor and orbit characteristics between the two missions.

Like SEVIRI, AVHRR uses three channels with similar, but slightly shifted central wavelengths and band widths (see Table 1). As for MDAL, broadband albedos are calculated based on albedo estimates made for all three channels. ETAL has recently been validated against MODIS collection 6 surface albedo by Lellouch et al. (2020), who concluded that the data meet the

accuracy targets set by the LSA-SAF, with a mean MBE of 0.001 and RMSD of 0.014 for albedos of less than 0.15, and a relative mean MBE and RMSD of 6 % and 19 %, respectively, for albedos greater than 0.15. Based on these findings we deem ETAL a suitable reference dataset in this study.

Following the acquisition of ETAL data through the LSA-SAF website (for the validation time period from November 2020 to October 2021), we resample them onto the MDAL grid through interpolation by inverse distance weighting (see example of

resulting albedo map in Fig. 3, right hand side).

### 3.3 Reference data 2: In-situ data

We compare MDAL v1 and v2 to in-situ observations of various stations in Europe and Africa: Cabauw, Evora, Gobabeb and Izaña; additional details for stations are listed in Table 2 and locations within SEVIRI disk are shown in Figure 1. Cabauw, Gobabeb and Izaña are part of the BSRN network (Driemel et al., 2018), Evora is operated by the Karlsruhe Institute of

Technology within the framework of the LSA-SAF. All stations measure shortwave downwelling (SWD) and upwelling (SWU) radiation, and have a data acquisition frequency of 1 min. For the three BSRN stations we use the same time period as for the comparison to ETAL, i.e. a full year November 2020 to October 2021 and we obtained the data from the BSRN data portal (https://dataportals.pangaea.de/bsrn/). For Evora we only use data from November 2020 to May 2021 due to technical problems at the site, the data was provided by the Karlsruhe Institute of Technology. These sites encompass all the stations within the

SEVIRI 0° disk that we found suitable for comparison, with data coverage for the validation period, and that measure both SWD and SWU.




| | Station Properties | | | | Prop. of nearest SEVIRI pixel | | |
|---|---|---|---|---|---|---|---|
| Station | Coordinates | Footprint diameter | Derived albedo $(\beta_{min/max})$ | Vegetation type | Pixel center coordinate | Footprint size | Vegetation type(s) |
| Cabauw, Netherlands (Knap, 2021) | E 4.9271 N 51.9684 | 46 m | white-sky $(\beta_{min} = 0.99)$ | Grassland | E 4.9062 N 51.9827 | Lon.: 3.2 km Lat.: 6.4 km | Grassland 57.3 % Savannas 37.3 % Cropland / Natural Vegetation Mosaics 4.0 % |
| Evora, Portugal | W 8.0033 N 38.5403 | <49 m | blue-sky | (Woody) Savanna | W 8.0100 N 38.5539 | Lon.: 3.2 km Lat.: 4.4 km | Savannas: 47.3 % Grasslands: 33.8 % Croplands: 16.2 % |
| Gobabeb, Namibia (Vogt, 2021) | E 15.0832 S 23.5195 | 46 m | black-sky $(\beta_{max} = 0.1)$ | Barren | E 15.0799 S 23.5136 | Lon.: 3.2 km Lat.: 3.5 km | Barren: 100 % |
| Izaña, Canary Islands (Cuevas-Agulló, 2021) | W 16.4991 N 28.3093 | 46 m | black-sky $(\beta_{max} = 0.1)$ | Barren / Open shrublands | W 16.4898 N 28.3088 | Lon.: 3.3 km Lat.: 3.8 km | Grasslands 70.8 % Open Shrublands: 22.9 % Woody Savannas: 4.2 % |

**Table 2.** In-situ stations used for comparison with MDAL and properties of nearest SEVIRI pixel. In-situ footprint was calculated using Equation 1, in-situ vegetation is inferred from visual inspection of high resolution satellite images (https://earthexplorer.usgs.gov). SEVIRI footprint vegetation types based on MODIS landcover MCD12Q1 (Section 3.4); it should be noted that the automatic classification algorithm may not always match "ground truth" (e.g. Savanna classification at Cabauw). SEVIRI footprint size based on distance between center coordinates of neighboring pixels.

Albedo values for each station can be calculated by taking the mean ratio of SWU to SWD. This yields *blue-sky albedo*, which is albedo observed under mixed illumination conditions, essentially falling somewhere between the two end-member cases of black-sky albedo (purely direct illumination conditions) and white-sky albedo (purely diffuse illumination conditions; Schaepman-Strub et al., 2006). Since MDAL albedo products are of either of the two latter types (i.e. **not** blue-sky), it is required to align in-situ albedo estimates with their satellite counterparts. Kharbouche et al. (2019) outline a procedure to estimate either white-sky or black-sky albedo from in-situ observations, which can be done if diffuse radiation is measured at the site. Essentially, this approach assumes that blue-sky albedo estimates from observations with very high diffuse radiation content are suitable proxies for white-sky albedo, while those with very low diffuse radiation content correspond to black-sky albedo. The method can be summarized in two steps:

1. Calculate the ratio $\beta$ of diffuse radiation to SWD.

2. For estimating white-sky albedo only consider time slots where $\beta$ tends towards 1 (i.e. greater than a chosen threshold value, e.g. 0.99), for black-sky albedo this ratio should tend toward 0 (i.e. less than a chosen threshold value, e.g. 0.1).

Whether we can estimate black- or white-sky albedo for a given station depends on the local atmospheric, as well as solar- and view angle, conditions during the investigated time interval; some stations will yield more observations with $\beta$ close to 1 while for others it may more often be close to 0. For any given station we will calculate one or the other (see Table 2) depending on how many valid data points are available for estimating either albedo type. Furthermore, there is a trade-off between the chosen



threshold value and the number of valid albedo estimates that can be generated; if the threshold is very close to 1 or 0, we can produce more reliable estimates for white/black sky albedo but at the same time only very few measurements may meet this

criterium. In order to obtain a sufficient number of data points we adjust the threshold value. We estimate white-sky albedo for Cabauw, black-sky albedo for Gobabeb and Izaña, and (due to lack of diffuse radiation measurements) blue-sky albedo for Evora (threshold values for white-sky and black-sky albedo estimates are given in Table 2).

### 3.3.1   Station footprint diameters & representativeness

The spatial representativeness of a station ("the degree to which a ground-based retrieval of surface albedo is able to resolve

the surrounding landscape extending to the satellite footprint", Román et al., 2009) depends on the difference in reflective properties between in-situ and satellite footprint. A key factor affecting this difference is, necessarily, the difference in scale of the observed surface patch, which is why the footprint of each in-situ station — i.e., the area within its field of view — needs to be known.

For an instrument measuring upward radiation, the diameter $D$ of the covered ground footprint is a function of the instrument's

field of view ($FOV$), the height of the instrument above the surface $h_{inst}$ and the local vegetation height $h_{veg}$ (Kharbouche et al., 2019):

$$D = \tan\left(\frac{FOV}{2}\right) \times (h_{inst} - h_{veg}). \tag{1}$$

For the stations Cabauw, Gobabeb and Izaña we have $FOV = 170°$ and mounting heights of $h_{inst} = 2\,\mathrm{m}$ with negligible $h_{veg}$, resulting in $d \approx 46\,\mathrm{m}$ (Table 2). In the case of Evora the instrument has a height of $h_{inst} = 13\,\mathrm{m}$, which is the height of the

tree canopies in the sparsely vegetated forest the station is located in, and has a field-of-view of $FOV = 150°$. Because the vegetation is sparse, the effective footprint is irregular and varies with viewing direction, depending on the distance and height of vegetation at a given azimuthal viewing angle. We give a maximum value in Table 2 calculated for a clear field of view.

The implications of the calculated footprint diameters for comparison to MDAL are further discussed in Section 4.2.

### 3.4   Auxiliary data: MODIS IGBP Landcover types

For the land cover type based analysis we use the MODIS *Land Cover Type Yearly L3 Global 500m Version 6* (MCD12Q1 v006) product of 2019 (Friedl and Sulla-Menashe, 2019), created using supervised classification of reflectance data (Sulla-Menashe and Friedl, 2018; Friedl et al., 2010). The dataset was obtained through the NASA Earthdata portal (https://earthdata.nasa.gov). We resample the land cover types from the 500 m MODIS grid onto the coarser SEVIRI grid by assigning each SEVIRI pixel the most common cover type of the MODIS pixels that are located within a 1.5 km radius around that pixel's centre (result

shown in Figure 4).



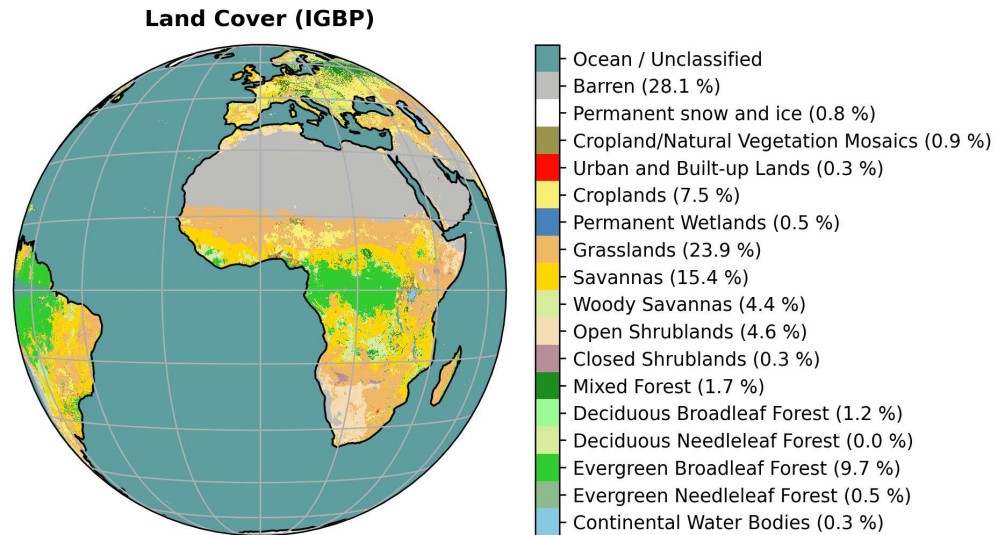

**Figure 4.** MODIS based IGBP land cover types, resampled on SEVIRI pixels. The description of each land cover type is given in the appendix, Table A1.

## 4 Validation strategy

Our validation is focused on an assessment of MDAL's performance with respect to the reference datasets presented in Section 2: ETAL and in-situ observations. To this end we calculate various error measures such as mean bias error (MBE), mean absolute error (MAE), root-mean-square deviation (RMSD), temporal correlation (see Appendix A4) and histograms based on

albedo maps and time series (see Section 5).

The basis of this validation is formed by the comparison between MDAL v1 and v2 values and the reference values taken from ETAL and various in-situ stations. Naturally, the differences between assessed data and reference vary both spatially and temporally which needs to be taken into account for the final assessment. For both types of reference data (ETAL and in-situ) we can conduct a temporary analysis with temporally average error measures, while for ETAL we can additionally run a spatial

analysis, over the full dataset, subregions and different land cover types. Further details about the procedure specific to either type of reference data are given below.

### 4.1 Validation against ETAL

While satellite data validation using other satellite based products suffers from the drawback that none of the compared products can claim to be "ground truth", it is still very useful, mainly because comparisons can be made with extensive spatial coverage

which can not be reached with in-situ data.





A successful comparison of satellite data sets requires spatial, temporal and spectral agreement between them. We obtain spatial agreement by interpolation of the more finely resolved ETAL product onto the grid of the coarser MDAL, as mentioned in Section 3.2. To satisfy spectral agreement, we perform the comparison of MDAL and ETAL only for broadband products (AL-BB-BH and AL-BB-DH) which allows us to ignore the slight differences in central wavelength and width for individual

channels (see Table 1). For temporal agreement of we need to consider that MDAL is a daily product while ETAL is a 20-day composite product, disseminated in ∼10 day intervals. The values for each composite are based on a Kalman filter which assigns weights to included observations based on their age (see Carrer et al., 2018; Geiger et al., 2008, for algorithm details), i.e. weights are highest on the most recent day and decrease exponentially for older observations. Hence, we follow the approach of Lellouch et al. (2020) and match MDAL and ETAL datasets at the last day of the ETAL composite (the 5th, 15th and 25th

of each month). An example of the matched datasets for one date can be seen in Figure 3.

For our analysis we match pixels of the respective MDAL versions with those of ETAL for each time slot and calculate various validation measures: bivariate histograms, MBE, MAE and temporal correlation coefficients. For additional detail we conduct in-depth analysis of temporal and spatial subsets, defined by specific season, region and land cover type. The land cover types are based on the IGBP classification (see Section 3.4); the sub-regions are: Eurasia, North Africa, Sub-Saharan Africa and

South America (see Figure 1).

## 4.2 Validation against in-situ data

In-situ observations are generally considered a good validation reference for satellite based albedo. The key advantage of in-situ measurements is the instrument's location close to the ground, much less affected by atmospheric effects than a satellite measurement. Because the covered area of in-situ and satellite observations are usually not identical, in particular with respect

to scale, the CEOS (Committee on Earth Observation Satellites) land product validation group recommends that "only sites that are spatially representative of the satellite field of view should be used for validation purposes" (Wang et al., 2019). While SEVIRI pixels have an extent of at least 3 km (at sub-nadir point, and increasing towards the edge of the disk; the size for each pixel covering an in-situ station is given in Table 2), the footprints of the in-situ stations used for reference in this study are generally just around 50 m (see values in Table 2; Figure 5 for visual reference), which is why the interpretation of the

difference between satellite and in-situ data needs to be done with care. To give proper context when presenting the results (Section 3.3), we will give a brief qualitative assessment by comparing the land cover at the station with the dominant land cover types within the MDAL pixel which we use to compare to the in-situ data. It should be noted, however, that a qualitative land cover comparison can only be a rough indication of the quality of the comparison of albedos as it does not capture the effect of the difference in scale very well, nor possible effects due to varying albedo within a single land cover type.

With this caveat in mind, we present the results of comparison between in-situ albedo and both MDAL version, for the SEVIRI pixel that covers the station, in Section 5.2. The type of albedo (BH or DH) we compare, depends on which type yields more data points (in turn depending on prevalent sky conditions, as well as view angles) for each of the stations (see Section 3.3).





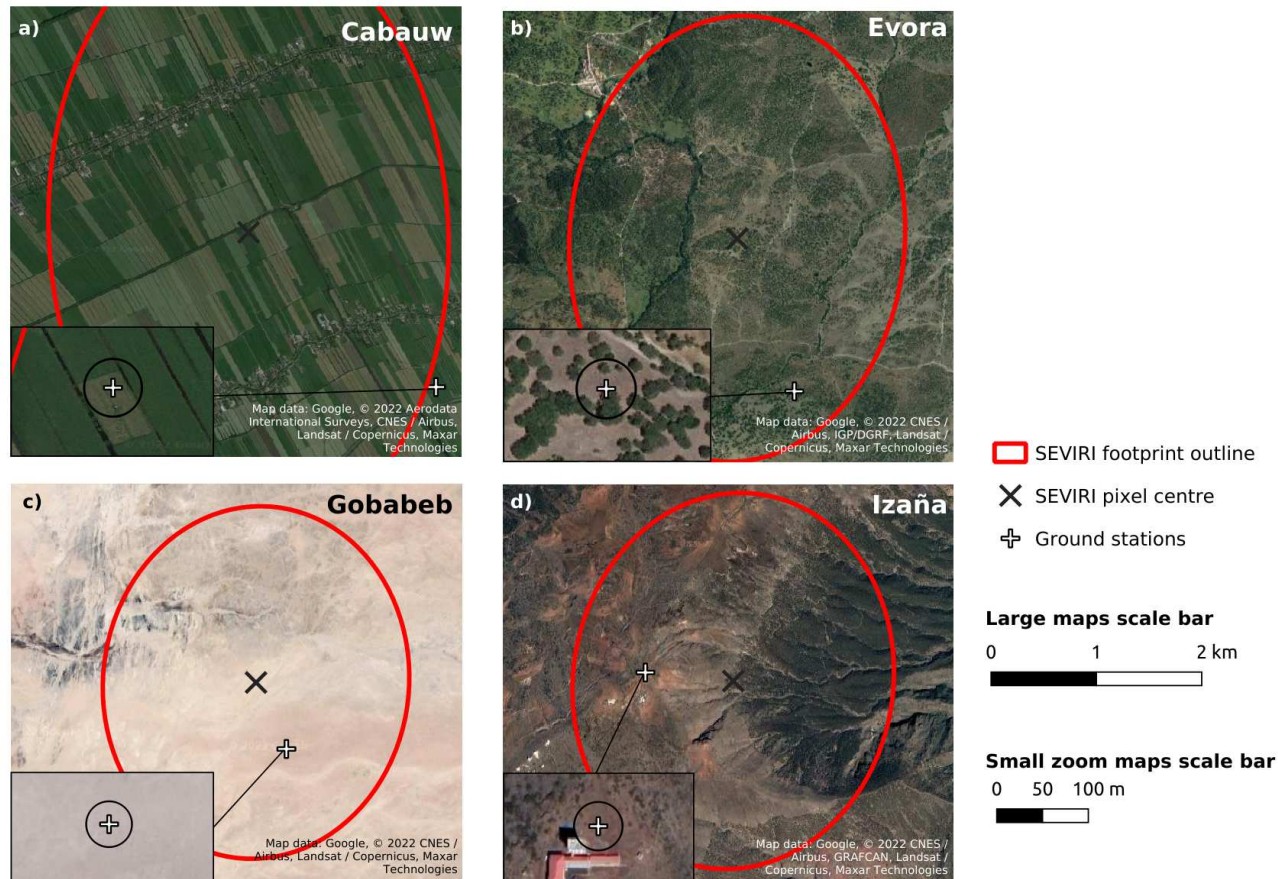

**Figure 5.** The footprints and surroundings of the in-situ stations Cabauw, Evora, Gobabeb and Izaña, as well as approximate footprints of the nearest SEVIRI pixel.

## 5    Results and discussion

### 5.1    Validation against ETAL

Figure 6 shows a comparison between both MDAL versions and ETAL (for white-sky albedo), based on mean error measures (MBE and MAE) calculated over the full validation time window (one full year). The MBE maps indicate a widespread under-estimation of albedo values in MDAL v1, which is notably reduced in v2 (on average from 0.02 to less than 0.001 below ETAL values). The same analysis for black-sky albedo (AL-BB-DH) reveals a slight average improvement from approximately $-0.02$ to $+0.01$ (Figure A4, top left), i.e. from negative to a noticeable but less pronounced positive bias (see Figure A1, top row). The MAE maps (Figure 6, second row) show good agreement between both MDAL versions and ETAL, although in the edge regions the high SEVIRI view zenith angles lead to less accurate results. The map also shows slightly elevated MAE just east





of the sub-satellite point along the coast of the Gulf of Guinea. It should be noted that this region exhibited worse than average performance in the validation of ETAL against MODIS (see figures 15 and 17 in Lellouch et al., 2020), a possible indicator that ETAL is less reliable here. Looking at averages over the full disk, however, there is an overall improvement of absolute errors from MDAL v1 to v2 w.r.t. ETAL, with a reduction from 0.034 to 0.026, an improvement of 24 % (for AL-BB-DH the change is from 0.028 to 0.024, or 14 % improvement, see Figure A5, top left). Temporal correlation by measure of Pearson's coefficient $r$ is shown in Figure 6, bottom row, where we can see an overall improvement, in particular in northern and central Africa. The poor correlations in the tropics obtained for MDAL v1, which prevail to lesser extent in v2 as well, are likely associated with cloud contamination. This introduces noise in both ETAL and MDAL estimates, which in an area where albedo is very stable, leads to negligible (or event negative) correlations between the different datasets.

Figure 7 shows a qualitative comparison in terms of which of the MDAL versions is on average closer to the ETAL value, for AL-BB-BH, highlighting that MDAL v2 is on average closer to ETAL than v1 for 76 % of pixels (66 % for AL-BB-DH, see Figure A2).

These observations are further corroborated by the 2-D histograms in Figure 8 (showing AL-BB-BH; see Figure A3 for AL-BB-DH), calculated for four different dates across all seasons, for pixels in the northern hemispheres (to ensure seasonal effects are not masked). The plots show a trend of negative bias in MDAL v1 for all dates (except January for AL-BB-DH) when compared to ETAL, which is eliminated in MDAL v2 resulting in improved correlation and errors w.r.t. ETAL except for January. A slight positive bias of MDAL v2 compared to ETAL, however, can be seen for albedo values greater than $0.4$ for most dates and both types of albedo.

For a more detailed analysis we additionally calculate mean MBE and MAE values for varying combinations of spatial (Eurasia, Northern Africa, Sub-Saharan Africa, South America) and temporal (individual seasons) subsets; values for MDAL v1 and v2 shown side by side in Figures 9 and 10 (first row in each). The MBE numbers in Figure 9 highlight that underestimation of albedos in MDAL v1 (w.r.t. ETAL) is prevalent in all regions and for all seasons. Similar to the average over the full disk dataset, significant improvements can be seen in MDAL v2 in mean MBE for all regions as well as notable improvements in mean MAE, most pronounced in northern Africa. There is only a single instance where MDAL v2 shows inferior error measures compared to v1: for Eurasia mean winter MAE increases slightly from 0.064 to 0.066. For black-sky albedo the seasonal dependence on the results is more obvious: In northern Africa and southern America the performance of MDAL v2 with respect to ETAL is diminished in local winter, but improved in local summer (Figures A4 & A5). In Eurasia there is a full year average deterioration of black-sky MDAL v2 performance compared to v1, both for MBE and MAE, and observable for almost all seasons except for MBE in spring. The magnitude of MBE and MAE differences varies between seasons but is again strongest in winter.

By distinguishing different land cover types, we attempt to determine whether certain cover types exhibit better or worse results than others, which may help explain the slightly diminished performance of MDAL v2 in winter. Indeed, Figure 11 shows MDAL v2 achieving smaller mean MAE values for all land cover types except inland water bodies, for all seasons except winter (neither MDAL nor ETAL are designed for retrieval of albedo of water surfaces, so this result is of no real concern).



**Figure 6.** Plots showing performance of MDAL v1 and v2 against ETAL for all pixels, for AL-BB-BH and analysis over one full year, from 1 Nov 2020 to 31 Oct 2021. Top: MBE; middle: MAE; bottom: Pearson's correlation coefficient R. Mean values of MBE and MAE are given in Figures 9 and 10.





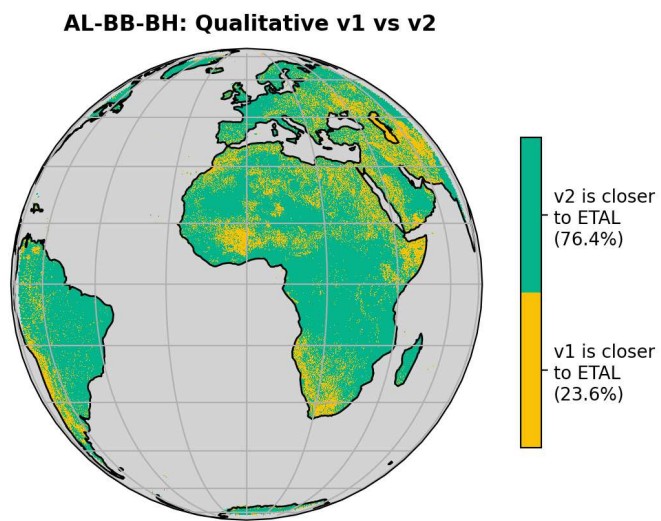

**Figure 7.** Qualitative comparison of performance of MDAL v1 and v2 with respect to ETAL, for broadband white-sky albedo (AL-BB-BH).

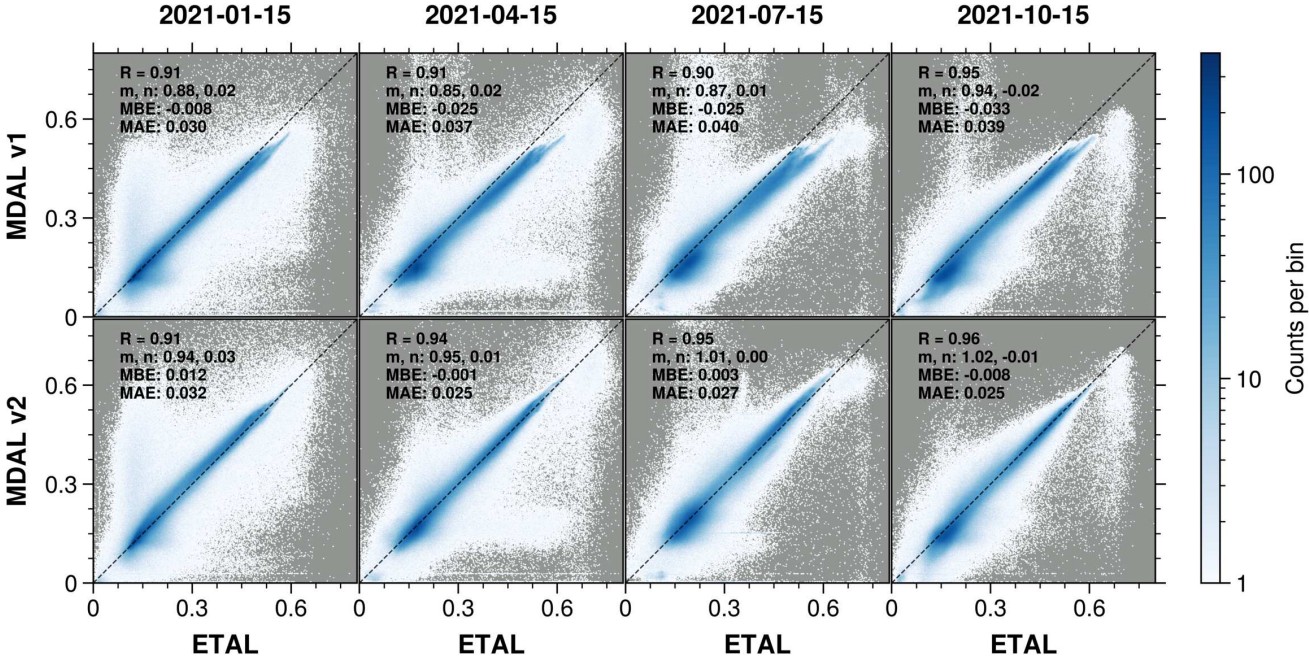

**Figure 8.** 2-D histogram highlighting correlation between MDAL v1 (left) and v2 (right), and ETAL, for BB-BH albedos for January, April, July and October 2021, 15th of each month. $R$ is Pearson's correlation coefficient, $m$ and $n$ are slope and intercept of a least-squares fitted linear regression line. Color scale is normalized to logarithmic scale for better visibility. Dashed line is the line of perfect correlation ($m = 1$, $n = 0$).


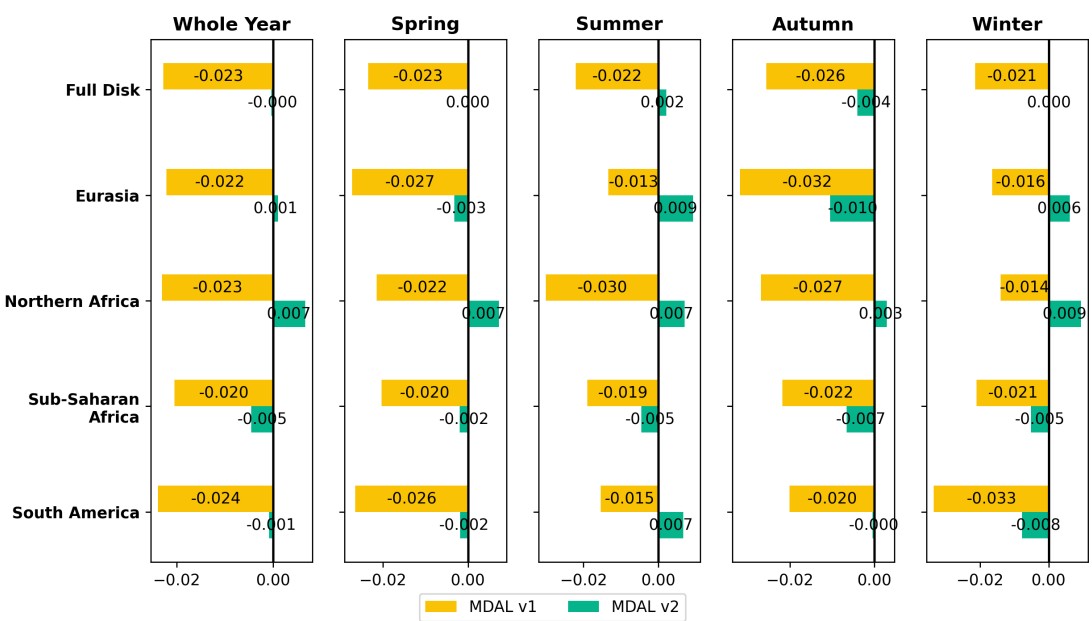

**Figure 9.** Mean MBE w.r.t ETAL by region and season (seasons names referring to season months of northern hemisphere: Spring: MAM; Summer: JJA; Autumn: SON; Winter: DJF).

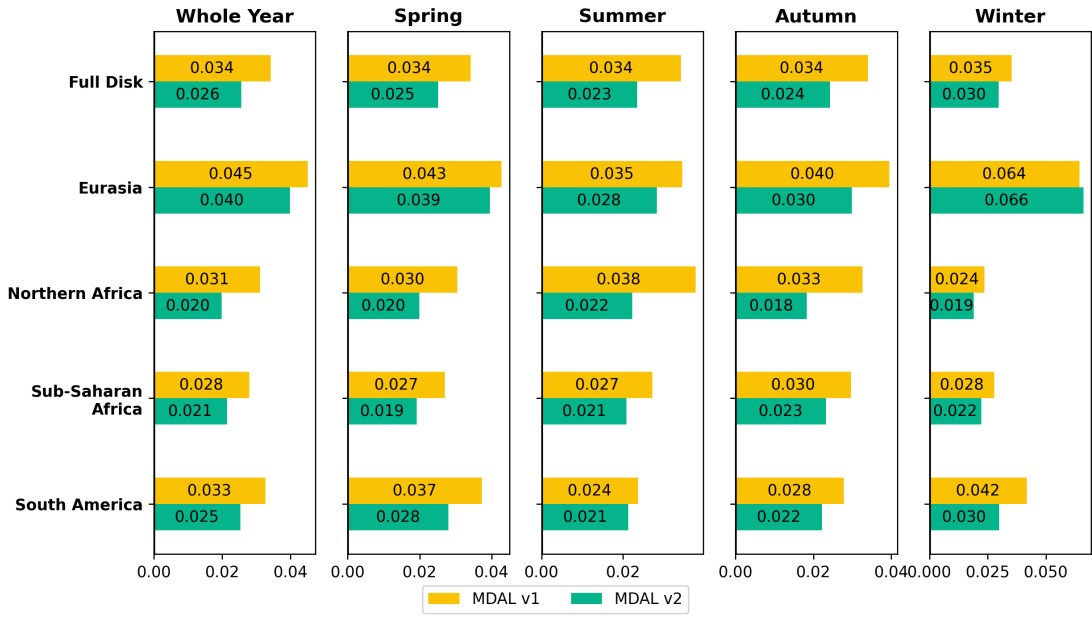

**Figure 10.** Mean MAE w.r.t ETAL by region and season (seasons names referring to season months of northern hemisphere: Spring: MAM; Summer: JJA; Autumn: SON; Winter: DJF).



For the other land cover types, we can see for winter that some are not improved, or show slightly increased absolute errors: these are crops, closed and open shrublands, grasslands and in Eurasia also 'barren' (but not globally). Crops and Grasslands are the most common land cover types in the Eurasia region (together around 50 % of pixels), which may contribute to the minor increase in MAE in winter in this region. Averaged over the whole disk (first column in Figure 11), however, we can

see that there is improvement in mean MAE for each land cover type. This can most strongly be seen for cropland/natural vegetation mosaics, (woody) savannas and barren land cover. The results for black-sky albedo (Figure A6), while showing worse results for MDAL v2 for barren land cover, deciduous broadleaf forests and urban- and built up lands in winter, as well as for evergreen needleleaf forests in summer, are generally similar to those of white-sky albedo and exhibit improvements for most land cover types.

### 5.2 Validation against in-situ

#### 5.2.1 Cabauw

Comparison between MDAL and Cabauw in-situ data for AL-BB-BH is shown in Figure 12a. The station is located on grass, while the SEVIRI footprint contains a mix of land covers (see Table 2): grasslands (57 %), savannas (37 %) and cropland/natural vegetation mosaic (4 %), according to the MODIS IGBP land cover product (the identification of savannas at this location may

be a flaw of the classification algorithm — the only trees in this area are those planted in nearby villages and along roads, see Figure 5a). We find that both MDAL versions are generally below the in-situ estimates, albeit v2 significantly less so with MBE changing from $-0.08$ in v1 to $-0.03$ in v2. In particular in winter and autumn MDAL increases noticeably from v1 to v2 (with a change of close to 0.1), becoming closer to the in-situ values. A very noticeable difference can be observed in February where a snow episode lead to very high in-situ values, which is only partly captured, or rather, smoothed, by MDAL.

This is caused by the Kalman filter implementation for the retrieval of MDAL which can lead to temporal smoothing when land cover changes suddenly (e.g.: after snowfall). RMSD is relatively poor for both MDAL versions with 0.09 in v1 and 0.07 in v2. The temporal correlation between MDAL and in-situ albedo is very low and does not change significantly in the updated MDAL version ($\sim 0.1$ for both versions), although this might be due to the lack of seasonal variations in the data at this location, making the correlation coefficient more susceptible to noise. The discrepancy between both MDAL versions and

in-situ measurement derived albedos is likely connected to the limited comparability between the two, due to the differences in scale and land cover described earlier. That is why, while the improvements in error measures such as MBE and RMSD may indicate an improvement in MDAL v2 relative to v1, this can not be stated with certainty.

#### 5.2.2 Evora

Due to lack of diffuse radiation measurements at Evora we compare the MDAL black-sky AL-BB-DH values to the in-situ

blue-sky albedo, for 7 months of data from November 2020 to May 2021 (Figure 12b). The tower-mounted sensor sees a mix of grassland and tree canopies (see Figure 5b), generally representative of a savanna type landscape while the satellite



**Figure 11.** Mean MAE w.r.t ETAL by landcover and season. For all pixels the "true" season was taken into account, depending on whether the pixel is located on the northern or southern hemisphere.





field-of-view covers a range of land covers, mostly savannas (47 %), grass- (34 %) and croplands (16 %; see Table 2). The fact that savannas here are essentially grass vegetation with a tree cover of 10-30 % (see Table A1) and the combination of savannas and grasslands adds up to 81 % can be seen as a positive for representativeness of the in-situ footprint (i.e. both

footprints comprising primarily of grass cover and tree canopies). However, the difference in scale between the two footprints in combination with the mixed vegetation necessarily implies that comparability remains problematic. That said, the results that can be seen in Figure 12b show that both MDAL versions agree well with in-situ albedo estimates with MDAL v2 being slightly closer with improvements in RMSD from 0.02 to 0.01, in MBE (from $-0.020$ to $+0.004$) and in temporal correlation (from 0.1 to 0.3). As for Cabauw, whether this indicates a real improvement of MDAL at this location can not be said with

certainty.

### 5.2.3  Gobabeb

We compare MDAL AL-BB-DH to the black-sky albedo estimate at Gobabeb (see Figure 12c). The station is located in an area of homogeneous land cover type, barren for both the in-situ instrument's footprint as well as SEVIRI's (see Table 2; Figure 5). We can see that both MDAL versions are in good agreement with the in-situ observations. In particular, the fit between MDAL

v1 and in-situ albedo at Gobabeb is good to begin with, with a RMSD value of 0.007 and MBE of $+0.003$. MDAL v2 estimates a higher albedo than v1 throughout the whole year with an MBE of $+0.020$, leading to a slightly worse fit with Gobabeb station (RMSD of 0.015), except for the months May to July where we see improvement. The high correlation coefficient of about 0.8 for both MDAL versions as well as the agreement of landcover type between station and satellite footprint (both being barren) indicate good representativity of the in-situ observation (although the latter does not rule out possible reflectance variations

*within* the barren cover). The slight deterioration of results at this location is consistent with the comparison to ETAL which finds smaller differences for MDAL v1 in this region (see Figure A2).

### 5.2.4  Izaña

The difference between the two MDAL versions for the pixel closest to Izaña is essentially negligible (see Figure 12d), and both are significantly lower than the albedo estimate from Izaña station. The higher in-situ estimate is probably due to a lack

of representativity of the local station footprint, which covers mainly bare ground on a mountain top; the SEVIRI footprint includes a significant amount of vegetation, in particular on the nearby mountain slopes (see Figure 5d), which likely contributes to the lower MDAL albedos.

Similarly to Cabauw, snow episodes in January and February 2021 are not fully captured. Again, this is in part caused by the incorporated Kalman filter, but also due the altitude difference between the in-situ station (2367 m, and on a mountain top) and

the average within the SEVIRI pixel.





**Figure 12.** MDAL v1 and v2 compared to albedo estimates at in-situ observation stations Cabauw, Evora, Gobabeb and Izaña.

# 6 Conclusions

We propose an upgrade of the retrieval algorithm behind the LSA-SAF MDAL product, to better account for physical conditions by inclusion of aerosol effects and to keep up with recent scientific developments (update of processing coefficients and sensor bias correction). The comparison of the MDAL albedo values pre- and post upgrade (v1 and v2) with respect to the reference

ETAL and in-situ data highlights significant changes in MDAL values. A previously noticeable negative bias in white-sky broadband albedo with respect to ETAL is now strongly reduced in most areas that are part of MSG primary coverage (and on average), while a similar negative bias in black-sky broadband albedo is changed, on average, to a smaller positive bias. The results imply that in regions where a negative bias in MDAL v1 exists (more common in white-sky albedo), MDAL v2 often offers clear improvements while in regions where agreement between MDAL v1 and reference data was good to

begin with (more common in black-sky albedo), there is a risk to obtain a positive bias in MDAL v2. On average, however, MBE, MAE and temporal correlation improve for both albedo types. In-depth analysis shows a certain dependence of the relative performance between MDAL v1 and v2 on season, region and land cover, highlighting slightly diminished results of v2 compared to v1 under certain circumstances, such as: increased white-sky albedo MAE in Eurasia in Winter, and increased black-sky albedo during the whole year; increase of black-sky albedo MAE in South America in autumn and winter and in

northern Africa in winter; introduction of positive biases in black-sky albedo in Eurasia, northern Africa and South America (all w.r.t. ETAL). Some land cover types show—almost exclusively in winter—a minor deterioration of results, such as crop-, grass- and shrublands. On average and for most regions, seasons and land cover types, however, the results for MDAL v2 are positive with general improvements w.r.t. ETAL by all applied error measures.

This general observed improved agreement between MDAL and ETAL is consistent with the nature of the update: now the

two products use the same aerosol inputs, narrow- to broadband conversion coefficients based on the same spectral database as well as the same base algorithm (6SV1) for calculating SMAC coefficients for atmospheric correction. This shows that, a), the implementation of the update was successful and the new version works as intended and, b), considering the recent favorable validation of ETAL against MODIS, that the changes in MDAL results can be seen as an improvement. On top of that, the increased alignment between MDAL and ETAL adds to the potential of exploiting synergies between the two products,

e.g. combining them to obtain increased data coverage for research purposes or for the generation of derived vegetation products (e.g. LSA-SAF SEVIRI and AVHRR based Leaf Area Index, LAI, and Fraction of Absorbed Photosynthetically Active Radiation, FAPAR, products).

The comparison to in-situ stations requires attention to comparability between the footprint of each respective station and the SEVIRI field-of-view at the station's location. Gobabeb has the most favourable ground situation in this regard, being very

homogeneous with a single land cover in both in-situ and satellite footprint (although due to the difference in spatial scale the two sensors likely do not see the same level of variation, even within a single land cover type). Here, we see good agreement of both MDAL versions with in-situ albedo, albeit with a slight increase in RMSD in v2 compared to v1. This slightly diminished performance of MDAL v2 is consistent with the comparison to ETAL, which shows a similar tendency in the region. For the other three locations the satellite captures additional land cover types that are not present in the field-of-view of the ground

station which poses a challenge for the comparison in these cases. For Cabauw and Evora, improvements in RMSD and MBE can be seen for MDAL v2, but we can not be certain whether these are *actual* improvements or merely artifacts, caused by a wanting representativeness of the in-situ footprint compared to the satellite one. At Izaña changes between MDAL v1 and v2 are negligible with significant difference to in-situ albedo for both. Problems with representativeness are most obvious at this location, with significant, visually noticeable land cover variability in the SEVIRI footprint.

Considering the sum of evidence gathered from the here presented validation exercise, the proposed update of the MDAL retrieval process (MDAL v2) is found to yield a veritably improved albedo product. Based on our findings we recommend the implementation of this update in the LSA-SAF MDAL operational near real-time processing chain. At the time of writing the MDAL v2 upgrade has already been implemented in the processing chain for the demonstrational IODC albedo product. Implementation of the upgrade in the processing chain for the MSG primary coverage product is awaiting the results of the

corresponding operational readiness review.

**Data Availability**

All data used in this study are freely available. Operational MDAL and ETAL products are accessible through the EUMETSAT LSA-SAF website (https://landsaf.ipma.pt/en/). Results of the MDAL v2 experiment are available at http://dx.doi.org/10.5281/zenodo.6414693. In-situ data used in this study are provided by third parties: BSRN and the Karlsruhe Institute of Technology.

**Author Contribution**

Retrieval algorithm update conceptualization by XC, IT; implementation by DJ. Validation methodology by DJ, XC; validation execution and visualization by DJ. Data acquisition and pre-processing by DJ. Processing of MDAL v2 experiment by SG, SF. Data acquisition and pre-processing by DJ. Writing by DJ (original draft and manuscript preparation), XC (review, editing), IT (review). Project administration by IT, XC.

**Competing interests**

The authors declare no conflict of interest.

**Acknowledgements**

MDAL & ETAL albedo data were provided by the EUMETSAT Satellite Application Facility on Land Surface Analysis (LSA-SAF: http://lsa-saf.eumetsat.int; Trigo et al., 2011). For help regarding in-situ data we thank: Frank Göttsche (Karlsruhe



Institute of Technology) for providing data and information for Evora and useful information for Gobabeb; Wouter Knap (Royal Netherlands Meteorological Institute) for providing additional information on Cabauw. This work was done within the framework of the LSA SAF (http://lsa-saf-eumetsat.int) project, funded by EUMETSAT.



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



**Appendix A**

**A1 Abbreviations**

**Appendix: Nomenclature**

| | |
|---|---|
| AL-BB-BH | Broadband white-sky (bi-hemispherical) Albedo |
| AL-BB-DH | Broadband black-sky (directional-hemispherical) Albedo |
| 520 AOD | Aerosol Optical Depth |
| AVHRR | Advanced very-high-resolution radiometer: Instrument aboard EPS-Metop satellites |
| BRDF | Bidirectional Reflectance Distribution Function |
| BSRN | Baseline Surface Radiation Network |
| CAMS | Copernicus Atmosphere Monitoring Service |
| 525 CEOS | Committee on Earth Observation Satellites |
| EPS | EUMETSAT Polar System |
| ETAL | EPS Ten Day Albedo |
| EUMETSAT | The European Organisation for the Exploitation of Meteorological Satellites |
| IGBP | International Geosphere-Biosphere Programme |
| 530 IODC | Indian Ocean Data Coverage |
| LSA-SAF | EUMETSAT Satellite Application Facility for Land Surface Analysis |
| MAE | Mean Absolute Error |
| MBE | Mean Bias Error |
| MDAL | MSG daily albedo |
| 535 MODIS | Moderate Resolution Imaging Spectroradiometer: Instrument aboard NASA's Terra and Aqua Satellites |
| MSG | Meteosat Second Generation |
| NWC SAF | EUMETSAT Satellite Application Facility for Nowcasting and Very Short Range Forecasting |





| RMSD | Root-mean-square deviation |
|------|---------------------------|
| SAF | EUMETSAT Satellite Application Facility |
| 540 SEVIRI | Spinning Enhanced Visible and InfraRed Imager: Instrument aboard Meteosat Second Generation satellites |
| SMAC | Simplified Method for Atmospheric Correction |
| SWD | Shortwave Downwelling (radiation) |
| SWU | Shortwave Upwelling (radiation) |
| TOC | Top-of-Canopy |

**A2   IGBP Land cover definitions**

**A3   Narrow- to broadband conversion**

MDAL broadband albedo estimates for a given target interval $\gamma$ are obtained by a linear combination of albedos calculated for the three utilized SEVIRI channels and a set of conversion coefficients $c_{\beta,\gamma}$ (Geiger et al., 2008):

$$a_\gamma = c_{0,\gamma} + \sum_\beta c_{\beta,\gamma} a_\beta \tag{A1}$$

The changes to $c_{\beta,\gamma}$ as part of the update of the MDAL retrieval algorithm described in this paper are given in Table A2.

**A4   Error measures**

**Mean bias error**

The mean bias error is the mean difference between two sets of observations, e.g. a set of values whose quality we want to assess $v$, and a set of reference values $v_r$.

$$\text{MBE} = \frac{1}{n} \sum_{i=1}^{n} (v - v_r) \tag{A2}$$



| Name | Description |
|---|---|
| Evergreen needleleaf forests | Dominated by evergreen conifer trees (canopy >2 m). Tree cover >60 %. |
| Evergreen broadleaf forests | Dominated by evergreen broadleaf and palmate trees (canopy >2 m). Tree cover >60 %. |
| Deciduous needleleaf forests | Dominated by deciduous needleleaf (larch) trees (canopy >2 m). Tree cover >60 %. |
| Deciduous broadleaf forests | Dominated by deciduous broadleaf trees (canopy >2 m). Tree cover >60 %. |
| Mixed forests | Dominated by neither deciduous nor evergreen (40-60 % of each) tree type (canopy >2m). Tree cover >60 %. |
| Closed shrublands | Dominated by woody perennials (1-2 m height) >60 % cover. |
| Open shrublands | Dominated by woody perennials (1-2 m height) 10-60 % cover. |
| Woody savannas | Tree cover 30-60% (canopy >2 m). |
| Savannas | Tree cover 10-30% (canopy >2 m). |
| Grasslands | Dominated by herbaceous annuals (<2 m). |
| Permanent wetlands | Permanently inundated lands with 30-60 % water cover and >10 % vegetated cover. |
| Croplands | At least 60 % of area is cultivated cropland. |
| Urban and built-up lands | At least 30 % impervious surface area including building materials, asphalt, and vehicles. |
| Cropland / natural vegetation mosaics | Mosaics of small-scale cultivation 40-60 % with natural tree, shrub, or herbaceous vegetation. |
| Snow and ice | At least 60% of area is covered by snow and ice for at least 10 months of the year. |
| Barren | At least 60% of area is non-vegetated barren (sand, rock, soil) areas with less than 10% vegetation. |
| Water bodies | At least 60% of area is covered by permanent water bodies. |

**Table A1.** IGBP land cover definitions, from Sulla-Menashe and Friedl (2018).

| Bandwidth interval $\gamma$ | $c_{0,\gamma}$ | $c_{1,\gamma}$ | $c_{2,\gamma}$ | $c_{3,\gamma}$ |
|---|---|---|---|---|
| [0.3 μm, 4 μm] | $0.004724 \rightarrow \mathbf{0.003600}$ | $0.5370 \rightarrow \mathbf{0.3563}$ | $0.2805 \rightarrow \mathbf{0.3596}$ | $0.1297 \rightarrow \mathbf{0.1496}$ |
| [0.4 μm, 0.7 μm] | $0.009283 \rightarrow \mathbf{-0.012500}$ | $0.9606 \rightarrow \mathbf{0.8153}$ | $0.0497 \rightarrow \mathbf{0.0838}$ | $-0.1245 \rightarrow \mathbf{-0.0815}$ |
| [0.7 μm, 4 μm] | $-0.000426 \rightarrow \mathbf{0.017400}$ | $0.1170 \rightarrow \mathbf{-0.0001}$ | $0.5100 \rightarrow \mathbf{0.5817}$ | $0.3971 \rightarrow \mathbf{0.3465}$ |

**Table A2.** Narrow- to broadband conversion coefficients used for MDAL albedo retrieval, before and after update. New values in bold.





**Mean absolute error**

The mean absolute error is the mean absolute difference between two sets of observations, e.g. a set of values whose quality we want to assess $v$, and a set of reference values $v_r$.

$$\text{MAE} = \frac{1}{n}\sum_{i=1}^{n} \mid v - v_r \mid \tag{A3}$$

**Root mean square deviation**

Root mean square deviation is the square root of the mean square error between a set of observations $v$, and a set of reference values $v_r$.

$$\text{RMSD} = \sqrt{\frac{1}{n}\sum_{i=1}^{n}(v - v_r)^2} \tag{A4}$$

**Pearson's correlation coefficient $R$**

The Pearson correlation coefficient measures the linear correlation between two data sets. For a set of observations $v$, and a set of reference values $v_r$, with respective means $\mu$ and $\mu_r$:

$$\text{R} = \frac{\sum_{i=1}^{n}(v - \mu)(v_r - \mu_r)}{\sqrt{\sum_{i=1}^{n}(v - \mu)^2 \sum_{i=1}^{n}(v_r - \mu_r)^2}} \tag{A5}$$

$R$ can range between $-1$ and $1$, where $1$ implies perfect linear correlation, $-1$ perfect anticorrelation and $0$ implies no linear dependency between the two variables.

**A5 Figures for black-sky albedo (AL-BB-DH)**





**Figure A1.** Plots showing performance of MDAL v1 and v2 against ETAL for all pixels, for AL-BB-DH and analysis over one full year, from 1 Nov 2020 to 31 Oct 2021. Top: MBE; middle: MAE; bottom: Pearson's correlation coefficient R. Mean values of MBE and MAE are given in Figures A4 and A5.

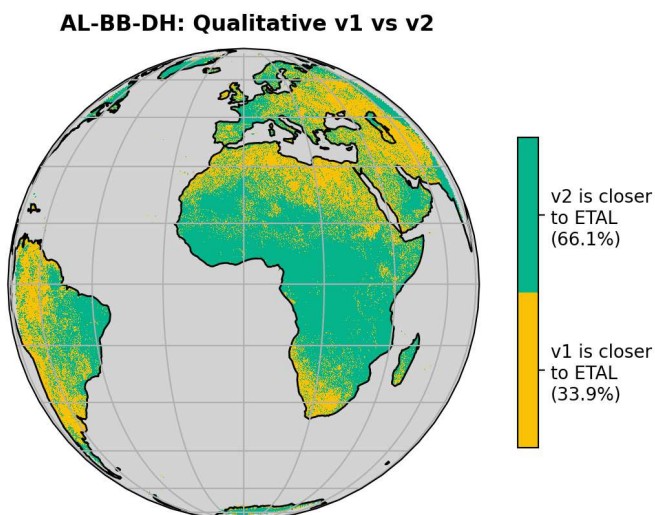

**Figure A2.** Qualitative comparison of performance of MDAL v1 and v2 with respect to ETAL for AL-BB-DH.

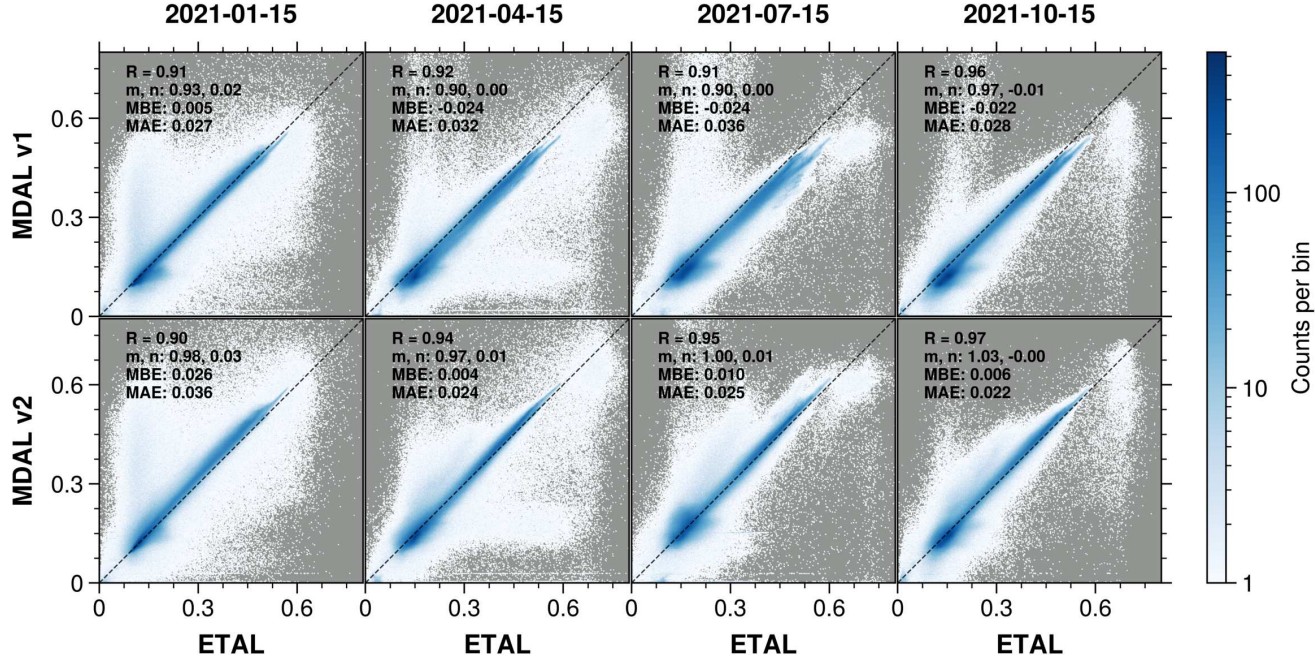

**Figure A3.** 2-D histogram highlighting correlation between MDAL v1 (left) and v2 (right), and ETAL, for BB-DH albedos for January, April, July and October 2021, 15th of each month. $R$ is Pearson's correlation coefficient, $m$ and $n$ are slope and intercept of a least-squares fitted linear regression line. Color scale is normalized to logarithmic scale for better visibility. Dashed line is the line of perfect correlation ($m = 1, n = 0$).





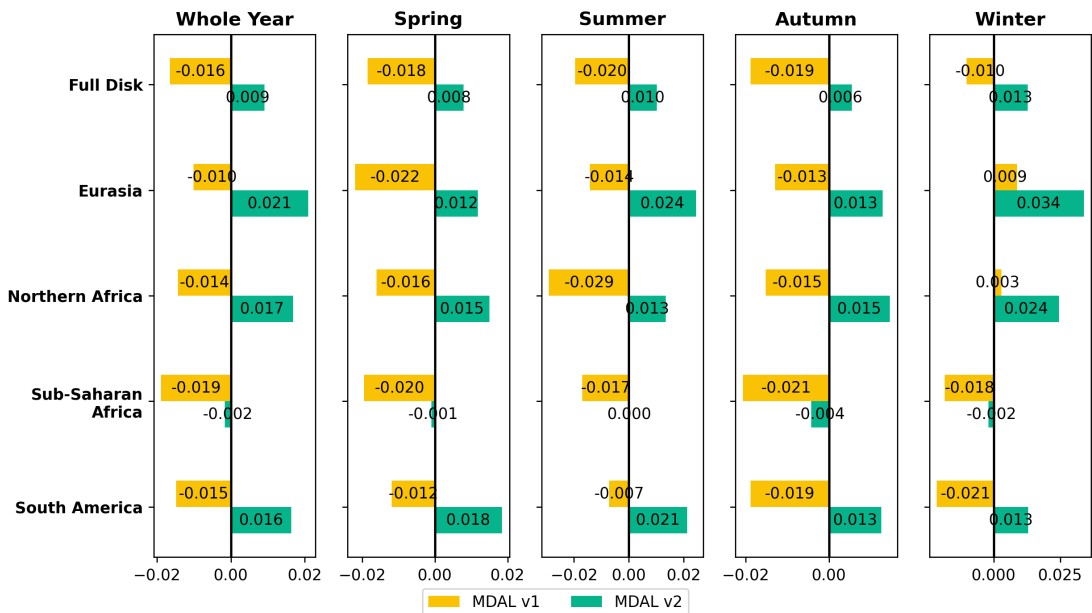

**Figure A4.** Mean MBE w.r.t ETAL by region and season (AL-BB-DH; seasons names referring to season months of northern hemisphere:
Spring: MAM; Summer: JJA; Autumn: SON; Winter: DJF).

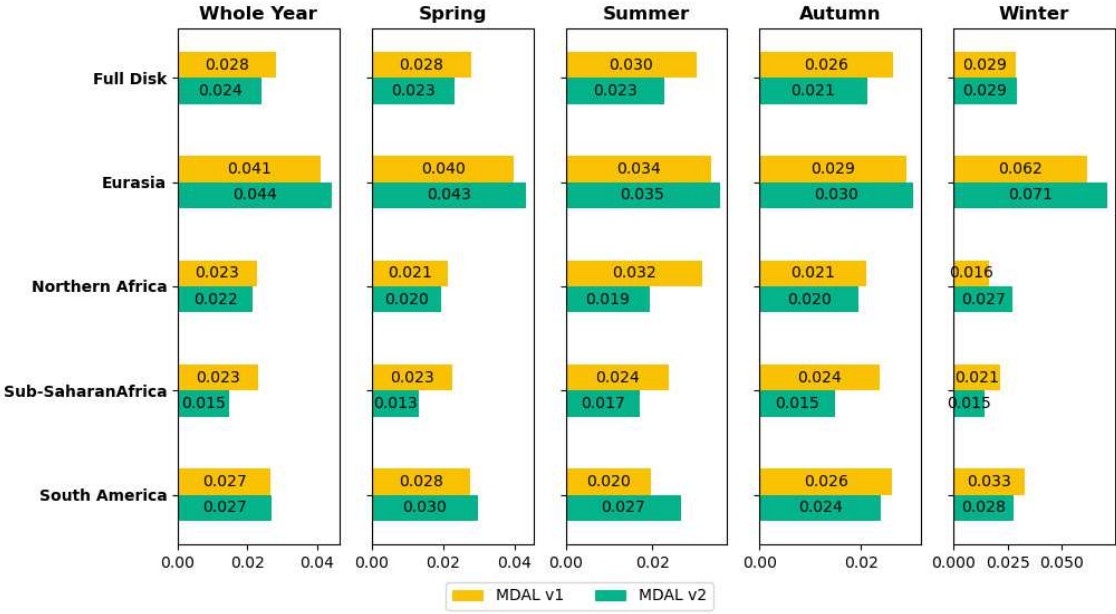

**Figure A5.** Mean MAE w.r.t ETAL by region and season (AL-BB-DH; seasons names referring to season months of northern hemisphere:
Spring: MAM; Summer: JJA; Autumn: SON; Winter: DJF).







**Figure A6.** Mean MAE w.r.t ETAL by landcover and season (AL-BB-DH). For all pixels the "true" season was taken into account, depending on whether the pixel is located on the northern or southern hemisphere.