# Peer review of "Upgrade of LSA-SAF Meteosat Second Generation daily surface albedo (MDAL) retrieval algorithm incorporating aerosol correction and other improvements"

_Geoscientific Instrumentation, Methods and Data Systems, 2022_

## Author Response (AR1)

**Replies to comments of reviewer 1**

We thank the reviewer for their time and the insightful comments on our manuscript. Below are our replies.

**Comment 1**
*67-68: As with the atmospheric correction, please cite the BRDF inversion algorithm here for easy reference to the reader.*

**Reply:**
We have added a reference to the BRDF model used.

**Comment 2**
*72-73: Advisable to refer here to the later more detailed description on NTBC improvements in section 2.2.*

**Reply:**
We have kept the sections on the current algorithm (2.1) and the updated one (2.2) separated on purpose for the sake of clarity. We do not deem this change necessary, since otherwise we would have to mention all the MDAL v2 changes in this bullet list (with some context), which would bloat the list. We have, however, added an additional reference for the NTBC methodology.

**Comment 3**
*92-96: One potential cause of the 'missing' additional bias due to missing aerosols may be that a large part of the SEVIRI disc is composed of medium-bright deserts like Sahara and the Arabian peninsula; over these targets whose surface albedo is often ~0.5, it has been shown that the presence of aerosols actually does quite little to alter the TOA-observable albedo – the target is neither bright or dark.*

**Reply:**
While the reviewer raises a valid point, this missing additional bias we are talking about appears globally and not only in regions of medium-bright surface albedo. This is what we are referring to in this paragraph.

**Comment 4:**
*134: What about heavy aerosol loading conditions of AOD550 > 1? SMAC would be expected to exhibit degraded performance in those conditions because of the internal parameterizations which increase its speed, would it not? Do you still process albedo under every possible AOD provided by the*

*reanalysis climatology? And are there plans to move from a climatological AOD to a dynamically updated one?*

**Reply:**
We are aware of this limitation of SMAC. Using an aerosol climatology somewhat alleviates this issue because the AOD values are less extreme. Furthermore, we attempt to avoid over-estimation of AOD by using the lower tercile (rather than the mean or the median) of 10 years of CAMSRA data. This point was not clear in the original manuscript, we have corrected this.

We have updated Figure 2 for the revised manuscript to show histograms of AOD for each month (new figure attached). In the added plots we can see that AOD values are usually well below 1, with a median value of <0.2 in every month.

There are currently no plans to change to a dynamically updated AOD for MSG/SEVIRI (but possibly for next generation MTG-I/FCI).

**Comment 5**
*135: The cutoff at SZA=80 already seems quite courageous, but what about View Zenith Angle? At the SEVIRI disc edge, the spatial footprint is very large and the atmospheric path lengths of the observed radiances are very long, which complicates the atmospheric correction considerably. Do you really retrieve albedos all the way to the disc edge?*

**Reply:**
Yes, MDAL retrieves albedos close to the disk edge and has done so since the product was first released. We agree with the reviewer's view that this can be problematic, in particular the combination of extreme solar angles with extreme view angles. This means that the small reduction in maximum SZA that we implemented in this update, from 85° to 80°, can help alleviate problems close to the disk edge. The maximum VZA has been kept at 85° in order to maintain high spatial coverage of albedo retrieval (satisfying the needs of LSA SAF Scandinavian users for example) as well as product continuity. Limitations of albedo retrieved at extreme geometries will be explained in the MDAL product documentation accompanying the release.

**Comment 6:**
*170: So, ETAL features an aerosol loading component in its atmospheric correction? Is the data source for that the same as for MDAL v2?*

**Reply:**

Yes, the data source is exactly the same. For MDAL the aerosol inputs are just resampled onto the SEVIRI grid. We have added this information to Section 2.2.

**Comment 7:**

*237: This may be semantical, but this reviewer considers inter-dataset analyses as "intercomparisons", because even MODIS is still an estimate of the true albedo, rather than a reference in itself.*

**Reply:**

With "validation" we do not mean to imply "against a true reference", as this is difficult to achieve for satellite-based observations of the Earth's surface (other satellites can not be a true reference, ground observations generally do not cover the same ground footprint). We have made a few changes to section headings to take the reviewer's comment into account, however.

**Replies to comments of reviewer 2**

We thank the reviewer for their time and the insightful comments on our manuscript. Below are our replies, the original comments by the reviewer are shown in italics.

**Major comments:**

*1. Only four in-situ sites were used for the validation. Three of them (Cabauw, Evora, and Izana) are obviously not spatially representative. For example, the Cabauw station is located in a small area of grassland surrounded by large areas of croplands (Figure 5). The authors also mentioned that the V2 improvement can not be said with certainty at these sites. Gobabet desert site is homogenous and both v1 and v2 MDAL albedo agree well with ground measurements. MDAL v1 performs a little better. Therefore, the in-situ data did not provide valuable information for MDAL v2 albedo evaluation.*

**Reply:** We agree that the comparison to in situ data only provides limited amount of information, with the exception of Gobabeb. We decided nonetheless to keep the other stations for sake of completeness and to give the most amount of information to the readers. We accompany the results with ample discussion on the caveats of using the in situ stations for comparison. Spatial representative issues are well known in the community working on validation of satellite land products but results using in situ data are generally considered to be interesting and necessary. We have also amended this part in the manuscript by additional satellite data, see next comment.

*2. The MDAL v2 albedo was also evaluated by the intercomparison with the ETAL albedo product. The results show good agreement between MDAL v2 and ETAL. While as the authors said these two products share the same retrieval algorithm and several ancillary input data (e.g. AOD). I'd suggest including independent satellite albedo products (at least one) for the intercomparison given that the in-situ validation effort of this manuscript is limited.*

**Reply:** We thank the reviewer for the suggestion and have, in order to complement the comparison of MDAL against in situ albedo, added MODIS albedo as an additional reference for the four ground locations. See updated Figure 12 as well as amendments in the text.

*3. I am confused about the aerosol for atmospheric correction. This manuscript claimed that the aerosol is lacking in MDAL v1 and incorporated in MDAL v2. However the MDAL algorithm changes record (https://landsaf.ipma.pt/en/products/albedo/albedo-copy/) showed that CAMS climatology aerosol has been integrated since 2020? Climatology monthly aerosol generated from early years (2003-2012) is applied in MDAL v2 albedo. How about the impact of this coarse temporal resolution and old aerosol on the accuracy of albedo retrieval? In particular recent years the aerosol loading varies due to the pandemic COVID-19 (https://www.sciencedirect.com/science/article/pii/S1352231020308621). More details of this are needed to be consistent.*

**Reply to first part of comment (the algorithm changes record on the LSA SAF portal):** The Algorithm Changes Record refers to a preliminary update of the MDAL product, which we now mention in the updated manuscript. The preliminary algorithm update contains a flawed implementation of accounting for aerosols which has been improved in the full update presented in this paper. We have now included reference to the preliminary update in the manuscript, see line 101.

**Reply to second part of comment (impact of temporal resolution of aerosol inputs):** Because the MDAL is a near-real time satellite product, our choices of aerosol data are essentially limited to long term averaged climatology, as we have used, or forecasts. The use of the former data was judged to be more robust, as aerosol forecasts may result in higher biases due to existing temporal shifts with respect to real aerosol conditions as it was observed in the past by our team. The decision of using an aerosol monthly climatology for the LSA-SAF albedo products (MDAL, ETAL) represents a significant improvement with respect to the previous situation in which no aerosol correction was done. In the future, studies may be carried out to assess the impact of using aerosol information at a higher resolution (e.g. model forecasts or satellite retrievals) when the quality of these data will be good enough to obtain improved values of surface albedo. Finally, we believe that a climatology generated based on data from 2003-2012 will provide better results than making no aerosol correction, despite the potential changes that aerosol load can have experienced in the past 10 years. Furthermore, changes in AOD at this temporal scale we found to be minor (lower than 0.05 decade$^{-1}$ in AOD; see Li *et al.*, ACP, 2014, https://acp.copernicus.org/articles/14/12271/2014/acp-14-12271-2014.html).

*4.* *The discussion of this manuscript is limited. There are some interesting results but without further discussion. For example. The MDAL v2 albedo is expected to be higher than v1 based on the updates listed. The albedo from sites Cabauw, Evora, Gobabeb confirmed it (Figure 12), but why the difference between MDAL v1 and v2 albedo is negligible at site Izaña? Figure 8 indicated that the improvement of MDAL v2 vs. ETAL is mainly over the high albedo values (> 0.5). I'd suggest adding some discussion of this.   Why is the difference between MDAL and ETAL quite different from AL-BB-DH and AL-BB-BH? The mean MBE of MDAL v2 AL-BB-BH and ETAL is less than 0.01 for all the 4 regions while AL-BB-DH showed large values, particularly Eurasia (Figure A4) that reach up to 0.034. The MAE of Eurasia is the highest compared to other regions. I wonder if the large view angles contribute to this (for both the albedo algorithm and aerosol effect)?*

**Reply:** We disagree with the assertion that the upgrade to v2 should cause albedo values for *all* pixels to be higher (this seems to be implied in this comment). While it is true for many regions, it is not true everywhere. This is because, for example, accounting for aerosols will lead to increased albedo for a given pixel only if the reflectance at this pixel is higher than that of the aerosols (and vice versa). Furthermore, while the introduction of the SEVIRI bias correction will indeed lead to increased albedo when taken by itself, the update

of SMAC and narrow- to broadband coefficients do not have such a one-sided effect. We also disagree that Figure 8 only shows improvement for v2 for high albedos. Figure 8 clearly shows improvement w.r.t ETAL for the whole range of albedo values with exception of maybe the data of 2021-01-15 (first column in this figure).

As for the large MAE pre- and post upgrade of MDAL, the reviewer raises a good point. We added this text highlighting this observation to the Results section, see line 363 and following lines.

**Minor comments:**

*Line 35: the authors referenced several papers that utilize the MDAL albedo product but the references to the albedo product itself are lacking. I'd recommend referencing the MDAL albedo papers (e.g. Geiger et al. 2008; Carrer, 2010, 2018, 2021; Lellouch, 2020) here.*

**Reply:** In this line we detail the use of albedo products in general, we do not talk specifically about our own products here. Hence we have cited the articles mentioned by the reviewer elsewhere.

*Figure 1: add the date of the SEVIRI image that was acquired.*

**Reply:** We have added the date in the updated manuscript.

*Line 200: the authors evaluated black and white sky albedo using in-situ measurements directly instead of generating blue sky albedo based on the ratio of diffuse radiation to SWD. The threshold of the ratio varies in order to obtain a sufficient number of data. Does the change of the threshold impact the evaluation?*

**Reply:** It does change in the sense that more tight threshold did not allow for a meaningful comparison. To the extent that it was possible to do a comparison with less data points, we did not observe significant differences that would have lead to different interpretations.